# TRPV4 activation triggers protective responses to bacterial lipopolysaccharides in airway epithelial cells

Yeranddy A. Alpizar[1,2], Brett Boonen[1,2], Alicia Sanchez[1,2], Carole Jung[3], Alejandro López-Requena[1,2], Robbe Naert[1,2], Brecht Steelant[4], Katrien Luyts[5], Cristina Plata[3], Vanessa De Vooght[5], Jeroen A.J. Vanoirbeek [5], Victor M. Meseguer[6], Thomas Voets[1,2], Julio L. Alvarez[1], Peter W. Hellings[4,7], Peter H.M. Hoet[5], Benoit Nemery[5], Miguel A. Valverde[3] & Karel Talavera[1,2]

Lipopolysaccharides (LPS), the major components of the wall of gram-negative bacteria, trigger powerful defensive responses in the airways via mechanisms thought to rely solely on the Toll-like receptor 4 (TLR4) immune pathway. Here we show that airway epithelial cells display an increase in intracellular $Ca^{2+}$ concentration within seconds of LPS application. This response occurs in a TLR4-independent manner, via activation of the transient receptor potential vanilloid 4 cation channel (TRPV4). We found that TRPV4 mediates immediate LPS-induced increases in ciliary beat frequency and the production of bactericidal nitric oxide. Upon LPS challenge TRPV4-deficient mice display exacerbated ventilatory changes and recruitment of polymorphonuclear leukocytes into the airways. We conclude that LPS-induced activation of TRPV4 triggers signaling mechanisms that operate faster and independently from the canonical TLR4 immune pathway, leading to immediate protective responses such as direct antimicrobial action, increase in airway clearance, and the regulation of the inflammatory innate immune reaction.

[1] Department of Cellular and Molecular Medicine, Laboratory for Ion Channel Research, KU Leuven, Leuven, 3000, Belgium. [2] VIB Center for Brain & Disease Research, 3000 Leuven, Belgium. [3] Department of Experimental and Health Sciences, Laboratory of Molecular Physiology and Channelopathies, Universitat Pompeu Fabra, Barcelona, 08003, Spain. [4] Department of Microbiology and Immunology, Laboratory of Clinical Immunology, KU Leuven, Leuven, 3000, Belgium. [5] Department of Public Health and Care, Laboratory of Environment and Health, KU Leuven, Leuven, 3000, Belgium. [6] Instituto de Neurociencias de Alicante, Universidad Miguel Hernández-CSIC, E-03550 San Juan de Alicante, Spain. [7] Department of Oto-Rhino-Laryngology, Upper Airways Research Laboratory, Ghent University, Ghent, 9000, Belgium. Correspondence and requests for materials should be addressed to K.T. (email: karel.talavera@kuleuven.vib.be)

Epithelial cells (EC) lining the airways constitute the fore-front line of defense against inhaled pathogens. Covered with a mucociliary layer and connected by tight junction proteins, EC serve as a structural barrier against inhaled pathogens, and control the screening of the luminal microenvironment by antigen-presenting cells[1]. Furthermore, EC are endowed with pathogen-recognition receptors such as Toll-like receptors (TLR), which detect microbial motifs and initiate events leading to the production of cytokines and recruitment of circulating leukocytes to the airways[2, 3]. In addition to its role in innate immunity, TLR4-dependent stimulation of EC has also been shown to modulate the activation, maturation, and migration of mucosal dendritic cells, thus playing a crucial role in antigen-specific immunity[4–6].

However, recent studies in sensory neurons showed that the recognition of pathogen-derived cues is not restricted to the classical immune-related receptors (e.g., TLRs, nucleotide-binding oligomerization-domain-like receptor, protease-activated receptors, and C-type lectin receptor). For instance, formylated peptides and α-hemolysin derived from gram-positive bacteria[7] and lipopolysaccharides (LPS) from gram-negative bacteria[8] directly activate nociceptive neurons. This leads to action potential firing that may underlie infection-associated pain sensations[7–10], and the release of the calcitonin gene-related peptide, which is implicated in neurogenic inflammation[8] and the regulation of immune responses[7]. Also airway ECs have been shown to detect molecules secreted by gram-negative bacteria. Activation of bitter taste receptors in these cells by bacterial quorum-sensing molecules induces an increase of intracellular $Ca^{2+}$ concentration, leading to nitric oxide (NO) production, augmented mucociliary clearance, and direct anti-bacterial effect[11]. Despite the crucial role of EC in defensive responses of the airways, it is currently unknown whether these cells are equipped with mechanisms allowing for fast, TLR-independent detection of bacterial endotoxins. We hypothesized that, like sensory neurons, EC may be able to quickly respond to acute stimulation with LPS through the activation of $Ca^{2+}$-permeable channels.

To test this, we determined the effects of LPS on intracellular $Ca^{2+}$ dynamics of airway EC. We found that LPS increases intracellular $Ca^{2+}$ concentration in mouse and human airway EC

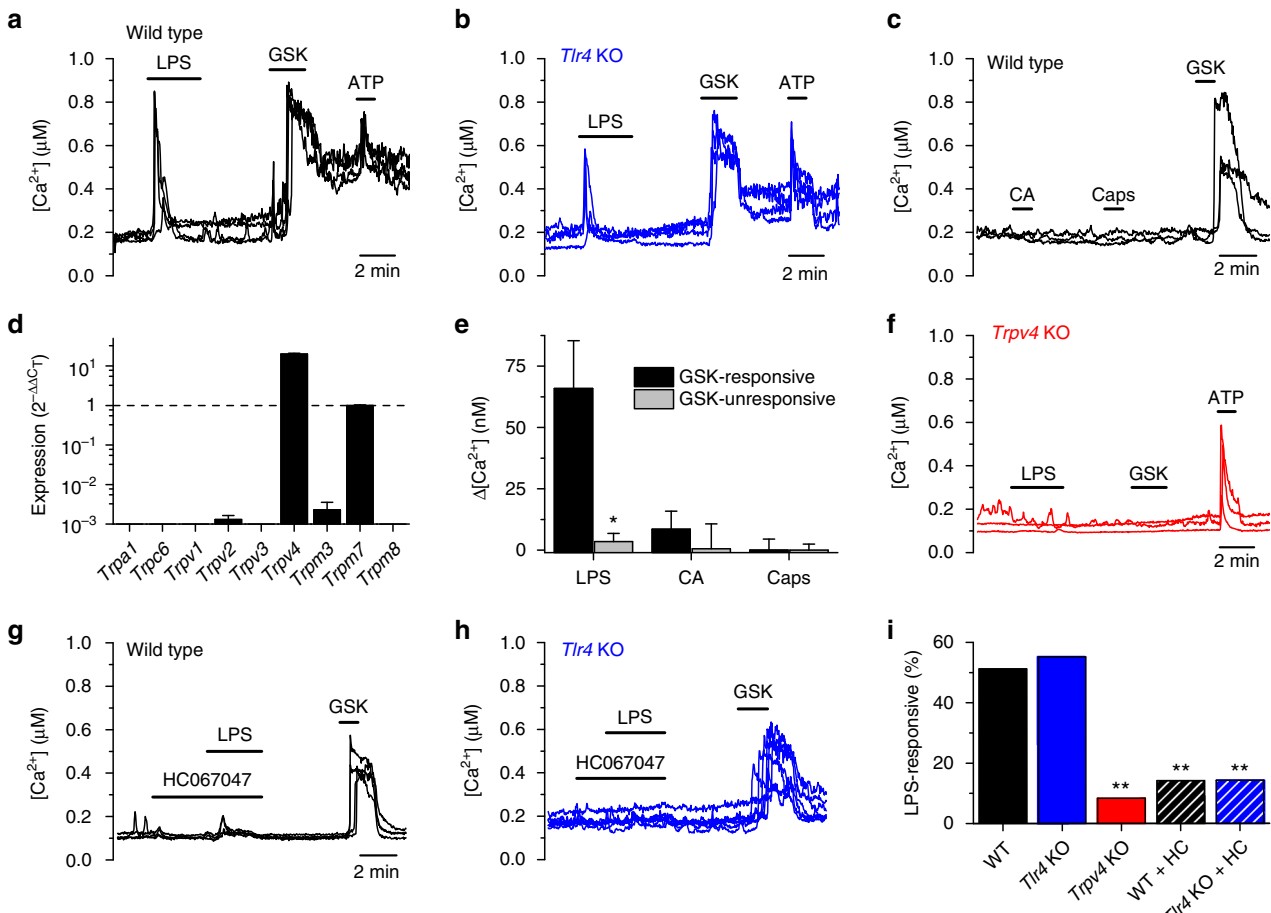

**Fig. 1** LPS stimulates mTEC in a TLR4-independent and TRPV4-dependent manner. **a**, **b** Effects of LPS (20 µg ml$^{-1}$), GSK1016790A (10 nM) and ATP (20 µM) on the intracellular $Ca^{2+}$ concentration in mTEC primary cultures from wild type (**a**) or *Tlr4* KO (**b**). **c** Traces of intracellular $Ca^{2+}$ concentration showing the lack of response of mTEC to the TRPA1 agonist cinnamaldehyde (CA, 300 µM) and to the TRPV1 agonist capsaicin (1 µM). **d** mRNA expression profile of several TRP genes. Data is shown as $2^{-\Delta\Delta CT}$ relative to *Trpm7* expression. **e** Average amplitudes of intracellular $Ca^{2+}$ responses to LPS (20 µg ml$^{-1}$), the TRPA1 agonist cinnamaldehyde (CA, 300 µM), and the TRPV1 agonist capsaicin (Caps, 1 µM) in cells responsive or irresponsive to 10 nM GSK1016790A. $n = 73$; *$P < 0.05$, Mann–Whitney $U$ test. **f** Effects of LPS (20 µg ml$^{-1}$), GSK1016790A (10 nM) and ATP (20 µM) on the intracellular $Ca^{2+}$ concentration in mTEC primary cultured from *Trpv4* KO mice. **g**, **h** Traces of intracellular $Ca^{2+}$ concentration in wild type (**g**) and *Tlr4* KO (**h**) cells exposed to 10 µM HC067047, 20 µg ml$^{-1}$ LPS and later on to 10 nM GSK1016790A (GSK). **i** Percentage of mTEC responding to LPS ($n > 46$). **$P < 0.01$, Fisher's exact test. HC, HC067047 (10 µM)

within seconds of application, via a mechanism that does not require TLR4, but the activation of the transient receptor potential (TRP) cation channel TRPV4. Furthermore, activation of TRPV4 by LPS triggers an increase in ciliary beat frequency (CBF) and an immediate release of NO, a compound that exerts direct antimicrobial and bronchodilation actions and that regulates neutrophil infiltration into the airways. In line with these results, the inhibition of TRPV4, either pharmacologically or genetically, results in enhanced ventilatory and inflammatory responses to LPS challenge in mice. Our data thus unveil TRPV4 as a key player in innate defense responses to bacterial endotoxins and argue against therapeutic inhibition of this channel in conditions associated with respiratory infections with gram-negative bacteria.

## Results

**TRPV4 mediates Ca²⁺ responses to LPS in airway EC**. We used intracellular $Ca^{2+}$ imaging to determine whether LPS acutely stimulates freshly isolated mouse tracheobronchial ECs (mTEC). We found an increase in intracellular $Ca^{2+}$ concentration in ~ 50% of the cells isolated from wild-type (WT) mice (Fig. 1a, i). Surprisingly, the prevalence of the responses was similar in cells isolated from *Tlr4* knockout (KO) mice (Fig. 1b, i), demonstrating a lack of requirement of TLR4-dependent signaling. The responses to LPS were virtually absent when $Ca^{2+}$ was omitted in the extracellular solution (Supplementary Fig. 1a, b), indicating

that they were primarily produced by $Ca^{2+}$ influx through the plasma membrane. LPS-induced responses were not mediated by either TRPA1 or TRPV1, two cation channels implicated in LPS effects in sensory neurons[8–10, 12], since we did not detect functional expression of these channels in mTEC (Fig. 1c–e). The responses to LPS occurred mainly in cells responding to the TRPV4 agonist GSK1016790A (Fig. 1e), suggesting an implication of this channel. Consistent with this idea, cells isolated from *Trpv4* KO mice were largely unresponsive to LPS and totally insensitive to GSK1016790A, but displayed normal responses to ATP (Fig. 1f, i). Furthermore, responses to LPS were drastically reduced in WT and *Tlr4* KO cells pre-incubated with the TRPV4 inhibitor HC067047 (Fig. 1g–i). Taken together, these data indicate that LPS induces $Ca^{2+}$ entry through TRPV4 channels in mTEC.

**LPS activates TRPV4**. To determine whether LPS activates TRPV4 channels we performed patch-clamp experiments in HEK293T cells transiently transfected with mouse TRPV4. Extracellular application of this compound elicited a fast and reversible increase in the amplitude of currents (Fig. 2a). Currents elicited by LPS showed typical features of TRPV4 activation, as their reversal potential and rectification pattern were similar to those of the currents activated by GSK1016790A (Fig. 2b–d). LPS was largely ineffective when applied in the presence of the TRPV4 antagonist HC067047 (Fig. 2e).

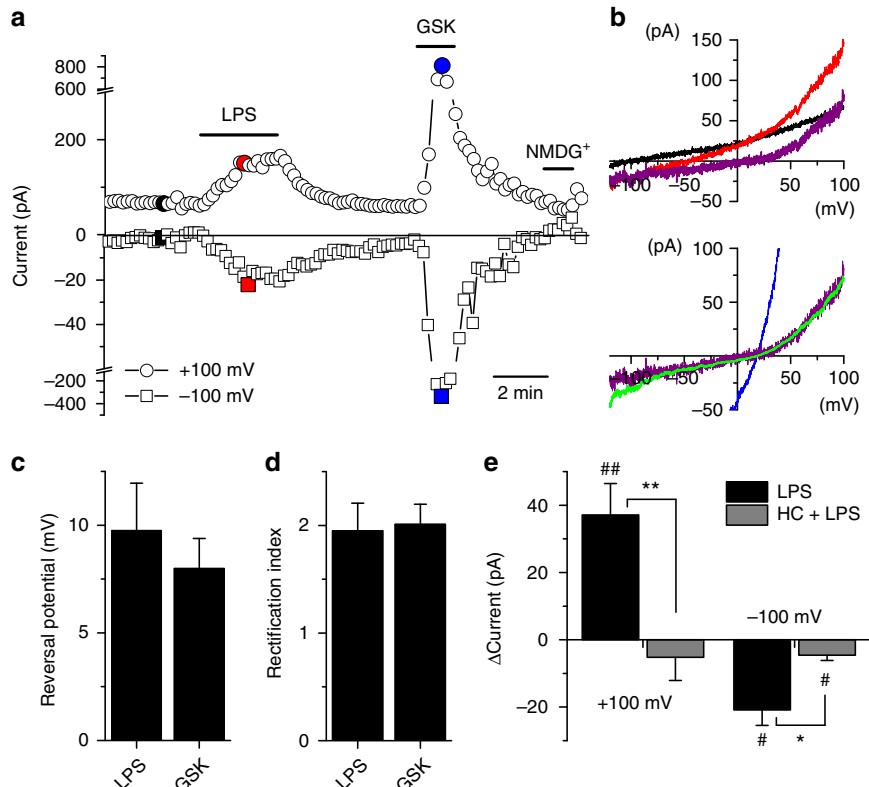

**Fig. 2** LPS activates TRPV4. **a** Example of the effects of LPS on the amplitude of currents measured at +100 and −100 mV in a HEK293T cell transfected with mTRPV4. The colored data points correspond to the traces shown in (**b**). **b** Top, currents recorded in control (black trace) or in the presence of LPS (20 μg ml⁻¹, red trace). The purple trace represents the difference of the red and black traces, i.e., the current induced by LPS. Bottom, comparison of the currents induced by LPS and GSK1016790A (10 nM, blue trace). The green trace represents the GSK1016790A-evoked current normalized to the amplitude of the LPS-evoked current measured at +100 mV. **c**, **d** Average reversal potential (**c**) and rectification index (**d**) of currents induced by application of LPS or GSK1016790A (n = 11). **e** Average change of current amplitudes induced by the application of 20 μg ml⁻¹ LPS (n = 11) or 20 μg ml⁻¹ LPS in the presence of 10 μM HC067047 (n = 7). # represents the comparison to ΔCurrent = 0 pA. #P < 0.05; ##P < 0.01 (n = 6), one sample t-test. *P < 0.05;
**P < 0.01 (n = 6), two-tailed t-test

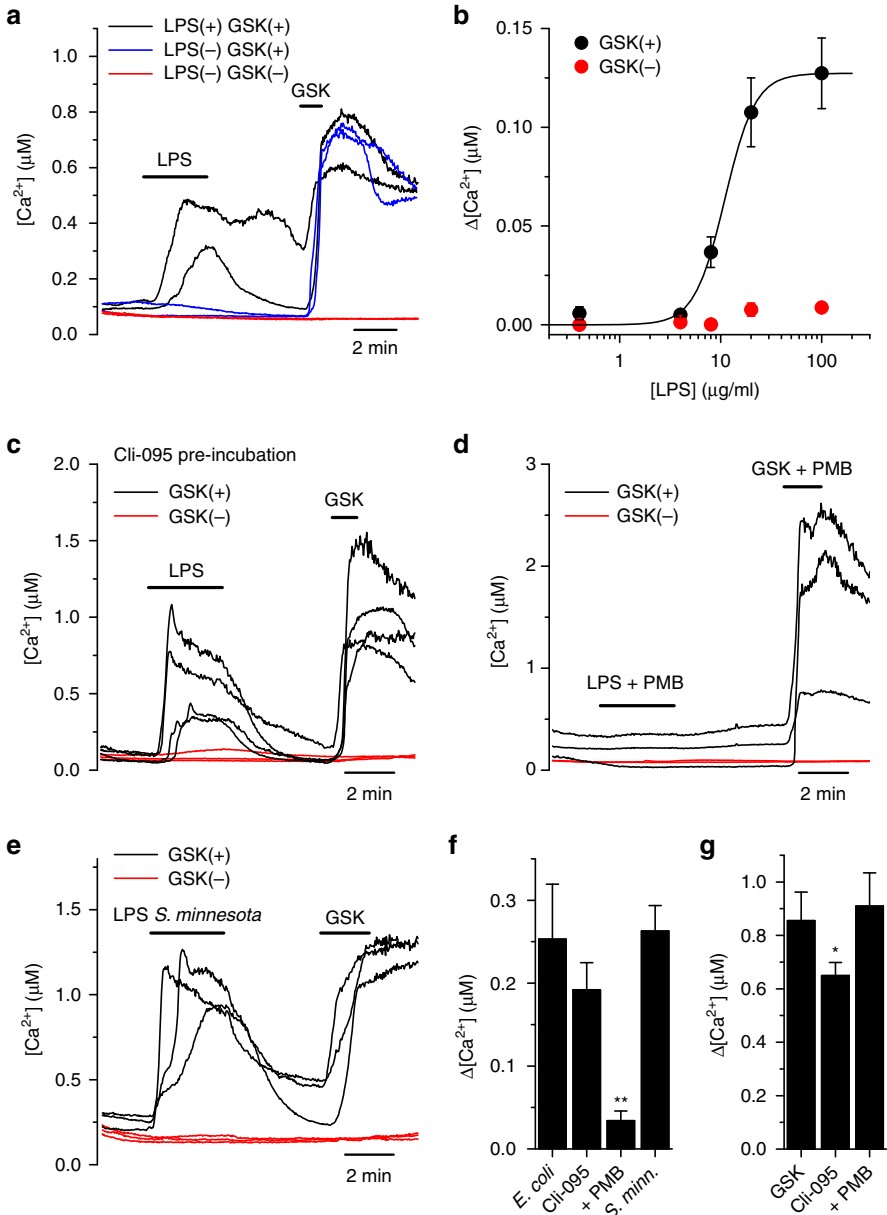

**Fig. 3** LPS activates TRPV4 in a TLR4-independent manner. **a** Intracellular Ca²⁺ signals in HEK293T cells overexpressing mouse TRPV4. The black and blue traces correspond to GSK1016790A-sensitive cells responding or not to 20 μg ml⁻¹ of LPS, respectively. The red traces correspond to non-transfected cells (GSK1016790A-insensitive). **b** Concentration dependence of the amplitude of LPS-induced intracellular Ca²⁺ responses ($n > 100$ per data point). The solid line represents the fit of the data obtained in GSK1016790A-positive cells with a Hill equation. The red symbols correspond to GSK1016790A-insensitive cells. **c** Intracellular Ca²⁺ signals recorded in HEK293T cells pre-incubated with the TLR4 antagonist Cli-095 (1 μM). **d** Intracellular Ca²⁺ signals recorded in HEK293T cells treated with LPS pre-incubated with Polymixin B (PMB, 300 μg ml⁻¹). In **c**, **d**, the black and red traces represent TRPV4-transfected and non-transfected cells, respectively. LPS 20 μg ml⁻¹ and GSK1016790A (GSK) 10 nM. **e** Intracellular Ca²⁺ signals in response to *Salmonella minnesota* LPS (20 μg ml⁻¹). **f** Average amplitude of the intracellular Ca²⁺ responses to 20 μg ml⁻¹ LPS from *E. coli* and *S. minnesota*, and to *E. coli* LPS in the presence of the TLR4 inhibitor Cli-095 (1 μM, $n = 77$) or the LPS scavenger Polymyxin B (PMB, 300 μg ml⁻¹, $n = 32$). **$P \leq 0.01$, two-tailed Kruskal–Wallis. **g** Average amplitudes of the responses to GSK1016790A (GSK) alone or mixed with Cli-095 or PMB in TRPV4-transfected HEK293T cells ($n > 30$ per bar). *$P < 0.05$, Dunn's multiple comparison test

As found in mTEC, LPS induced a robust increase in intracellular Ca²⁺ concentration in HEK293T cells heterologously expressing mouse TRPV4 (sensitive to GSK1016790A; Fig. 3a). This effect was concentration-dependent, with an effective concentration (EC₅₀) of $11.0 \pm 0.7$ μg ml⁻¹ and a Hill coefficient of $2.9 \pm 0.4$ (Fig. 3b). HEK293T cells not expressing TRPV4 (unresponsive to GSK1016790A) were virtually insensitive to LPS (Fig. 3a, b). LPS-induced responses in TRPV4-expressing cells were fully absent when Ca²⁺ was omitted in the extracellular

solution (Supplementary Fig. 2a, b) and strongly reduced by HC067047 (Supplementary Fig. 2c). Taken together, the patch-clamp and Ca²⁺ imaging data demonstrate that LPS can induce Ca²⁺ entry via the activation of TRPV4 channels.

Next, we wanted to confirm that the canonical TLR4 pathway is not required for the activation of TRPV4 by LPS. Pre-incubation with the TLR4 inhibitor Cli-095 did not affect the response of TRPV4-expressing cells to LPS (Fig. 3c, f) and had a marginal inhibitory effect on the response to GSK1016790A

(Fig. 3c, g). On the other hand, co-application of Polymyxin B (PMB), an LPS scavenger that binds to the lipid A moiety, strongly reduced the responses to LPS (Fig. 3d, f), but did not affect the response to GSK1016790A (Fig. 3d, g). Hence, LPS can activate TRPV4 independently of TLR4, and the lipid A moiety of LPS is required to be freely available for channel activation. We found that LPS extracted from *Salmonella minnesota* is also capable of activating TRPV4 (Fig. 3e, f), indicating that, unlike for TRPA1[8–10], the shape of lipid A moeity is not determinant in the stimulation of TRPV4. In the following we explored the cellular and systemic consequences of LPS-induced activation of TRPV4 in airway ECs.

**LPS-induced activation of TRPV4 triggers NO production**. Upon stimulation with LPS, airway ECs produce NO[13–15], a molecule that plays crucial roles in the mechanisms of host defense[16–18]. We hypothesized that LPS-induced TRPV4 activation may modulate NO concentration. NO production is typically evaluated several hours after the LPS challenge, but considering the fast effects of LPS on TRPV4, we were interested in a possible NO response during acute LPS stimulation. Thus, we performed simultaneous measurements of intracellular $Ca^{2+}$ concentration and NO levels. Application of LPS triggered increase in intracellular $Ca^{2+}$ concentration and subsequent NO production (Fig. 4a, e, f). mTEC derived from *Trpv4* KO mice did not produce NO upon LPS application (Fig. 4b, e, f). In WT mTEC, inhibition of TRPV4 with HC067047 significantly reduced the increase in intracellular $Ca^{2+}$ concentration and the NO production during stimulation with LPS (Fig. 4c, e, f). In contrast, NO was produced after LPS-induced intracellular $Ca^{2+}$ increase in cells isolated from *Tlr4* KO mice (Fig. 4d–f). Stimulation of both WT and *Tlr4* KO mTEC with GSK1016790A also induced an increase of NO (Fig. 4e, f and Supplementary Fig. 3c, b). The effects of LPS on intracellular $Ca^{2+}$ and NO production were recapitulated in human nasal ECs freshly isolated from inferior turbinate of healthy, non-smoking individuals (Supplementary Fig. 4a, b). These findings, together with the fact that the increase in NO signals occurred within 1 min after the raise in intracellular $Ca^{2+}$ concentration, are indicative of a fast $Ca^{2+}$-dependent mechanism of NO production.

NO is mainly produced from the oxidation of arginine by NO synthases (NOS), a process that requires the presence of $Ca^{2+}$/calmodulin coupling to the reductase domain of NOS[19]. Thus, we analyzed the presence of NOS isoforms in lysates from mTEC. We found that mTEC isolated from WT, *Tlr4* KO and *Trpv4* KO mice express the neuronal (n) NOS and the inducible (i) NOS (Fig. 4g). We were unable to detect the endothelial (e) NOS isoform (Supplementary Fig. 15a).

The fact that the nNOS isoform is directly activated by $Ca^{2+}$/CaM binding suggests that this isoform is a major contributor to the LPS-induced $Ca^{2+}$-dependent production of NO. Nonetheless, the constitutive expression of iNOS prompted us to determine whether the LPS-induced TRPV4 activation can regulate iNOS enzymatic activity. To evaluate this, we used a human bronchial EC line (16HBE), where we found iNOS to be the only NOS isoform constitutively expressed (Fig. 5a). The vast majority of these cells (96.5%) were responsive to GSK1016790A (Fig. 5b), demonstrating a wide functional expression of TRPV4. Application of 20 μg ml⁻¹ LPS increased the intracellular $Ca^{2+}$ concentration in 35% of cells (Fig. 5b) and HC067047 prevented LPS-induced responses in a reversible manner (Supplementary Fig. 5a). Experiments performed in HEK293T cells further confirmed that, like for the mouse isoform, LPS activates human TRPV4 (Supplementary Fig. 5b, c).

As observed in mTEC, the TRPV4-mediated $Ca^{2+}$ increase induced by LPS in 16HBE cells was followed by a rapid production of NO (Fig. 5d, e). Preventing the intracellular $Ca^{2+}$ increase by inhibiting TRPV4 with HC067047 significantly reduced NO production induced by LPS or GSK1016790A (Fig. 5d, e). Pre-incubation with the NOS inhibitor Nω-Nitro-L-arginine methyl ester (L-NAME, 1 mM) strongly reduced NO production without affecting the TRPV4-mediated intracellular $Ca^{2+}$ increase (Supplementary Fig. 6a, b), indicating that NO is mainly derived from iNOS activation.

We were able to discard other $Ca^{2+}$ entry pathways as possible contributors to the LPS-induced NO production. It has been reported that NO can be generated in skin keratinocytes from nitrite precursors by changes in intracellular pH induced through TRPV3-mediated $Ca^{2+}$ increase[20]. However, we found TRPV3 to be very poorly expressed in 16HBE cells (Supplementary Fig. 7a) and that transfection of this channel in HEK293T cells rendered these cells sensitive to the TRPV3 agonist camphor, but not to LPS (Supplementary Fig. 7b). Furthermore, we found that TRPA1 and TRPV1 are not functionally expressed in 16HBE cells (Supplementary Fig. 7c). We also confirmed that the ubiquitously expressed $Ca^{2+}/Mg^{2+}$-permeable channel TRPM7 was not stimulated by LPS (Supplementary Fig. 7d, e).

Interestingly, iNOS protein levels were not changed during the immediate response to LPS exposure (Fig. 5c), demonstrating that the NO production does not involve *de novo* synthesis of enzymes but a post-translational regulation. In rat aortic vascular smooth muscle cells iNOS has been shown to aggregate in inactive pools and to become functional upon disaggregation induced by activation of the $Ca^{2+}$/CaM pathway[21]. Similarly, we detected iNOS associated in aggresomes in 16HBE cells (Fig. 5f). Application of LPS or GSK1016790A reduced the number of iNOS aggresomes, an effect that was prevented by inhibition of TRPV4 with HC067047 (Fig. 5f, g). Taken together, these data indicate that activation of TRPV4 triggers disaggregation of iNOS.

**Anti-bacterial effect of TRPV4-mediated NO production**. To evaluate the anti-bacterial effect of EC-derived NO and the possible implication of TRPV4 we incubated non-pathogenic *E. coli* in exponential growing phase (0.1 optical density, OD) with confluent cultures of mTEC and quantified the percentage of dead bacteria using a live/dead (green/red) stain after 2 h. Incubation of bacterial cultures with WT mTEC killed ~ 13% of cells, whereas exposure to mTEC lacking TRPV4, or to cells pre-incubated with HC067047 or L-NAME strongly reduced the anti-bacterial effect (Fig. 6a, b). As control of these experiments we found that untreated cultures had $1.4 \pm 0.3\%$ dead cells and that the known bactericidal isopropanol induced $80 \pm 9\%$ cell death (Supplementary Fig. 8). We corroborated the ability of the 0.1 OD bacterial culture to induce intracellular $Ca^{2+}$ signals in TRPV4-transfected HEK293T cells (Fig. 6c, d). The percentage of responding cells was reduced in the presence of HC067047 or upon pre-incubation with PMB (Fig. 6c, d and Supplementary Fig. 9a, b). This strongly suggests that free LPS present in the bacterial culture is responsible for TRPV4 activation.

**LPS increases ciliary beat frequency**. In ciliated ECs, TRPV4-mediated $Ca^{2+}$ entry in response to changes in tonicity, fluid viscosity, or purinergic stimulation is crucial in the regulation of CBF[22–24]. Thus, we evaluated the effect of LPS-induced activation over beating frequency of ciliated mTEC. Application of LPS increased the CBF in mTEC cells isolated from WT mice, but failed to induce any effect on *Trpv4* KO cells or in WT cells pre-incubated with HC067047. The increase in CBF by LPS was

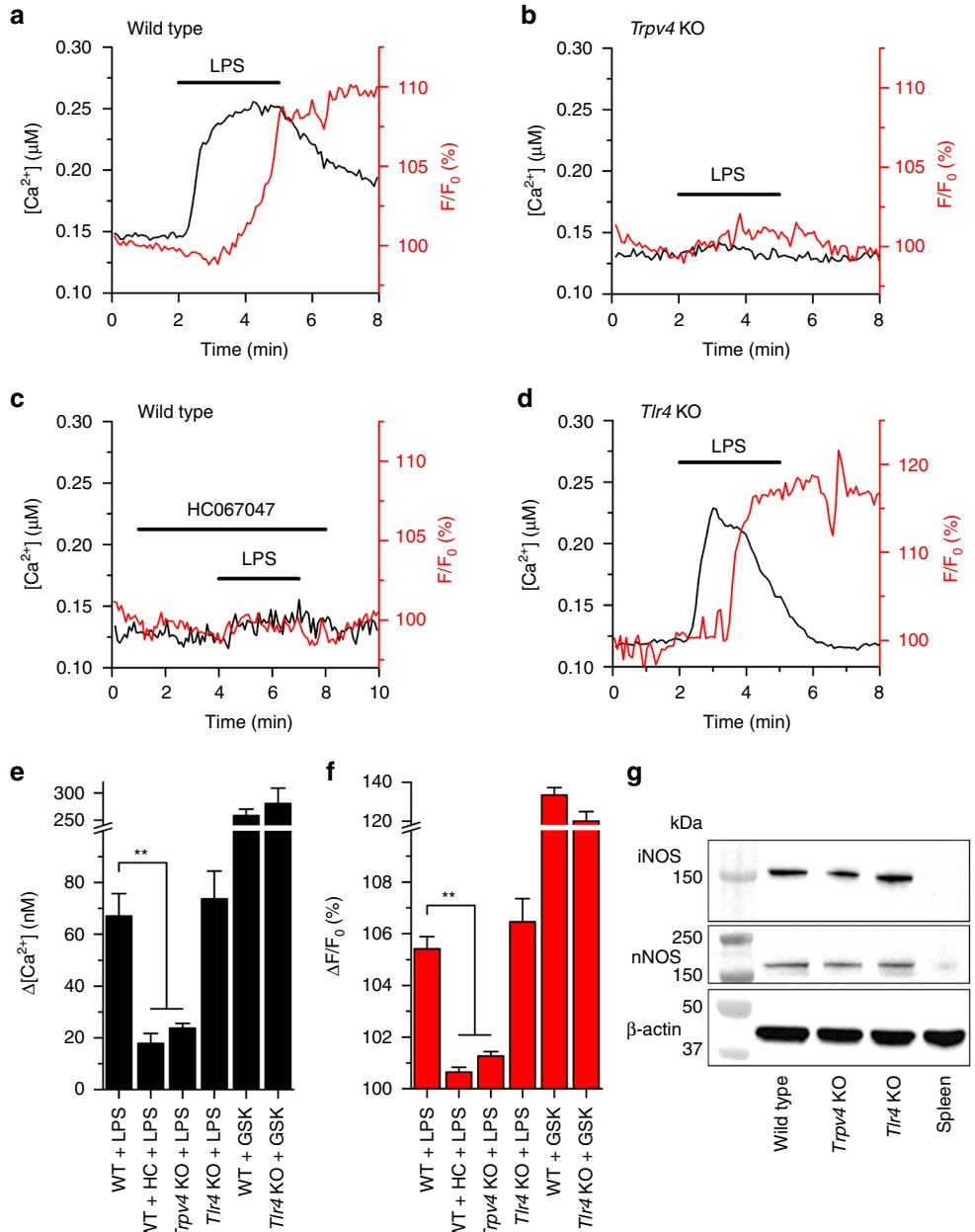

**Fig. 4** LPS-induced activation of TRPV4 triggers NO production in mTEC. **a–d** Traces of intracellular $Ca^{2+}$ signals (black traces) and normalized fluorescence of the NO-sensitive dye DAF-FM ($F/F_0$, red traces) recorded in mTEC harvested from wild type (**a**, **c**), *Trpv4* KO (**b**) and *Tlr4* KO (**d**) mice (LPS, 20 μg ml$^{-1}$; HC067047, 10 μM). **e**, **f** Average changes in intracellular $Ca^{2+}$ (**e**) and NO production (**f**) induced by LPS in wild type (WT), *Trpv4* KO and *Tlr4* KO. In another series of experiments, WT cells were also pre-incubated with the TRPV4 inhibitor HC067047 (HC, 10 μM). The two bars on the right show the responses to GSK1016790A (GSK, 10 nM) in WT and *Tlr4* KO mice. $n > 40$ per bar; **$P < 0.01$, Kruskal–Wallis test. **g** Western blot of mTEC lysates probed for the iNOS and nNOS isoforms. Non-stimulated splenocytes were used as negative control for NOS expression

not affected by L-NAME, excluding the implication of NO (Supplementary Fig. 10).

**TRPV4 regulates response to LPS in the airways.** To determine whether TRPV4 is involved in airway responses to LPS in vivo, we monitored the parameters describing the ventilatory function of mice using unrestrained whole-body plethysmography and assessed the numbers of lung-infiltrating macrophages and neutrophils in bronchoalveolar lavage fluids. We found that none of these magnitudes were significantly different between WT and *Trpv4* KO mice in control condition (Supplementary Fig. 11a–g),

indicating that TRPV4 is not a determinant of the basal ventilatory function or leukocyte infiltration into the airways. Administration of LPS, but not of control aerosols, significantly altered the ventilatory pattern of WT mice, including an increase in the enhanced pause (Penh, Fig. 7a and Supplementary Fig. 12a, b), in agreement with previous studies[25, 26]. LPS induced significantly stronger ventilatory changes (Fig. 7a, d) and cell infiltration responses (Fig. 7b, c, e, f) in *Trpv4* KO and in WT mice treated with HC067047. Although LPS induced changes in several ventilatory parameters, the increase in the time of expiration (Te) determined most of the difference of the response in Penh between WT and *Trpv4* KO mice (Supplementary Fig. 13a–f).

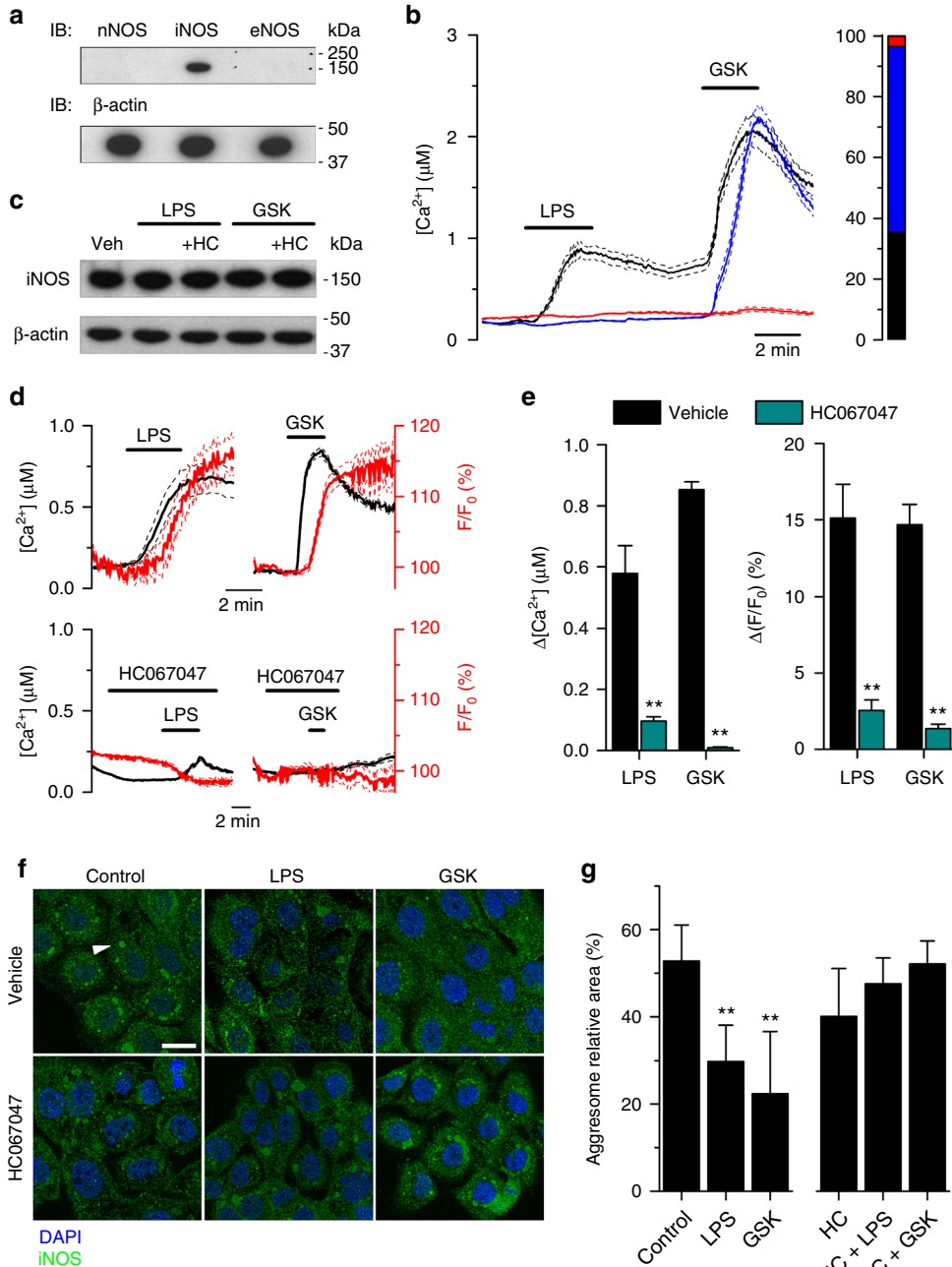

**Fig. 5** TRPV4 activation by LPS triggers NO production in human bronchial EC. **a** Immunoblotting of 16HBE cells lysate probed for NOS isoforms. **b** Left, average intracellular Ca²⁺ responses of human bronchial epithelial cells to 20 μg ml⁻¹ LPS and 10 nM GSK1016790A. The black, blue, and red solid traces correspond to data means of cells responding to LPS and GSK1016790A, GSK1016790A alone or to none of the stimuli, respectively. The thin dashed traces represent the means ± standard errors. Right, stack bar graph showing the percentages of cells responding to LPS and/or GSK1016790A (color-coded as the graph on the left; $n = 116$). **c** Immunoblotting of 16HBE cells lysate incubated in LPS (20 μg ml⁻¹) or GSK1016790A (10 nM). Where indicated, cells were previously incubated with vehicle (Veh, 0.8% DMSO) or the TRPV4 inhibitor HC067047 (+HC, 10 μM). **d** Average intracellular Ca²⁺ signals (black solid traces) and normalized fluorescence of the NO-sensitive dye DAF-FM (F/F₀, red solid traces) recorded in 16HBE cells ($n = 52$). The thin dashed traces represent the means ± standard errors. LPS, 20 μg ml⁻¹; HC067047, 10 μM; GSK1016790A, 10 nM. **e** Average changes in intracellular Ca²⁺ (left panel) and NO production (right panel) induced by LPS. The dark cyan bars correspond to cells pre-incubated with the TRPV4 inhibitor HC067047 (10 μM). $n > 74$ per bar; **$P < 0.01$, two-tailed $t$-test. **f** Confocal microscopy images of 16HBE cells immunostained for iNOS. The arrow head points to an aggresome structure. Scale bar, 20 μm. **g** iNOS aggresomes were quantified using a custom-designed software. The graph shows the area of aggresomes relative to the total green-stained area determined from images recorded from preparations fixed in control condition or 5 min after application of 20 μg ml⁻¹ LPS or 10 nM GSK1016790A. The panel on the right corresponds to cells pre-incubated with HC067047 (HC, 10 μM). Three randomly selected fields per condition were used for quantification. **$P ≤ 0.01$, Dunn's multiple comparison test

Next, we probed for possible contributions of the major airway chemosensors TRPA1 and TRPV1[44–46] in the responses to LPS. *Trpa1/Trpv1* double KO mice exhibited basal ventilatory properties that were similar to WT mice (Supplementary Fig. 11a–g). LPS induced a significantly larger increase in Penh than in WT mice (Fig. 7g), but no exacerbated neutrophil infiltration (Supplementary Fig. 14a, b). The ventilatory and cellular responses to LPS were significantly higher when TRPV4 was pharmacologically inhibited in *Trpa1/Trpv1*-deficient mice (Fig. 7g, and Supplementary Fig. 14a, b). Taken together, these findings demonstrate that TRPV4 is a main regulator of acute ventilatory and innate immune responses in the airways upon LPS challenge.

Finally, in line with the exacerbated neutrophil infiltration in LPS-challenged *Trpv4* KO, we found that mTEC derived from these mice produced significantly larger amount of neutrophil chemotractant *Cxcl-1* and interleukin (*Il*)-6 transcripts when acutely stimulated with LPS (Fig. 7h). Although LPS induced also the expression of *Cxcl-2*, its levels were similar in WT and *Trpv4* KO mice.

## Discussion

LPS is one of the most potent pathogen-associated signals for the immune system of vertebrates and it is generally believed to be detected solely by the TLR4 signaling pathway[27]. We found that LPS triggers an immediate response in intracellular $Ca^{2+}$ in airway ECs and that this occurred via a TLR4-independent mechanism. In addition, we found that TRPC6, a $Ca^{2+}$-permeable channel recently described to be crucial for the TLR4 signaling pathway in vascular endothelial cells[28], is not

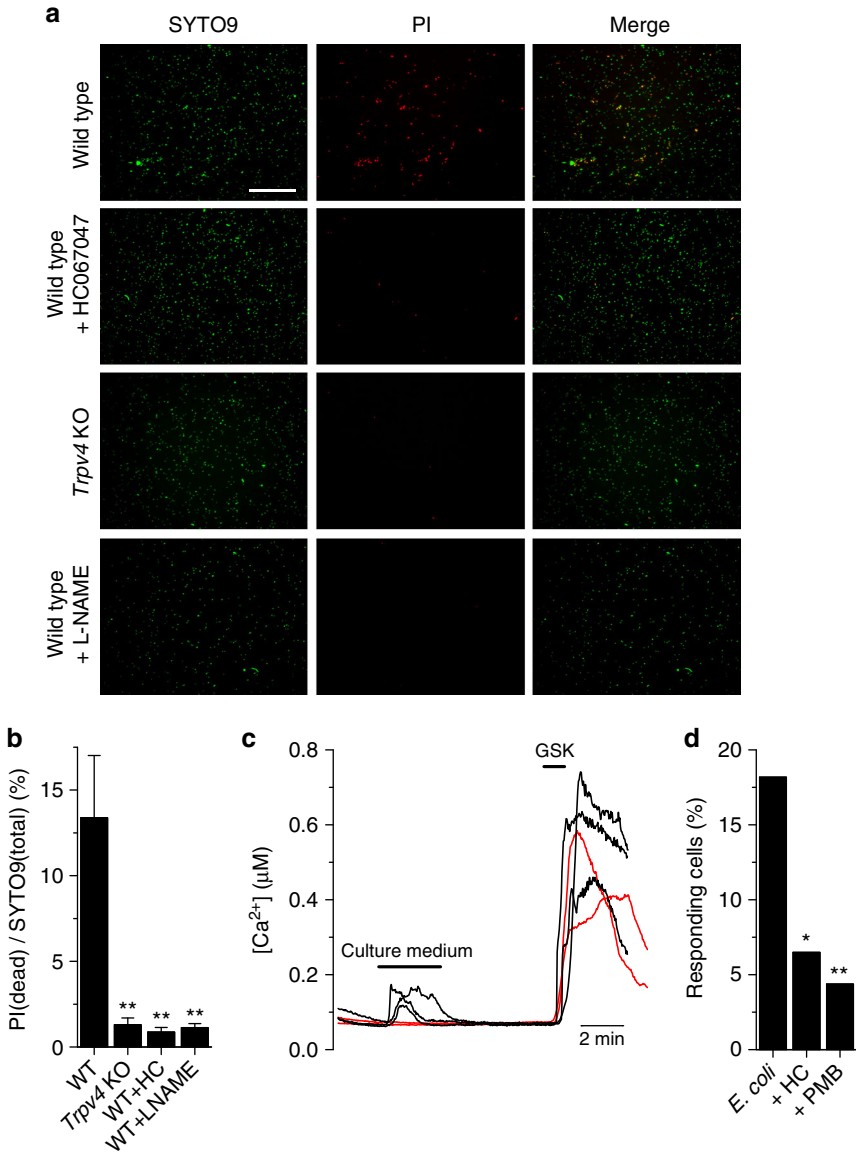

**Fig. 6** TRPV4-induced NO exerts direct anti-bacterial effect. **a** Representative images of *E. coli* bacteria stained with SYTO9 (green, all cells) or PI (red, dead cells). Magnification, ×40. Bacteria were imaged after 2 h incubation with a monolayer of mTEC isolated from WT or *Trpv4* KO mice. The TRPV4 antagonist HC067047 (10 μM) or the NOS inhibitor L-NAME (1 mM) were added 15 min prior to the bacterial challenge. Scale bar, 50 μm. **b** Percentage of dead cells (red/green) after co-culture with mTEC isolated from WT or *Trpv4* KO mice ($n \geq 5$). **c** Intracellular $Ca^{2+}$ response of TRPV4-transfected HEK239T cells to the supernatant of a 0.1 OD culture of *E. coli*. GSK1016790A, 10 nM. **d** Percentage of TRPV4-expressing HEK293T cells responding to supernatant of 0.1 OD culture of *E. coli*. The TRPV4 inhibitor HC067047 (HC, 10 μM) was perfused 2 min prior to addition of bacterial supernatant. The LPS scavenger Polymyxin B (PMB, 300 μg ml$^{-1}$) was pre-incubated during 30 min with the bacterial supernatant. *$P < 0.05$; **$P < 0.01$, Fisher's exact test

expressed in freshly isolated mTEC nor in the 16HBE cell line. We show here that, in contrast, LPS activates TRPV4, a channel that is prominently expressed in airway ECs. LPS elicited currents with characteristic signatures of activation of TRPV4, including a

reversal potential and a rectification pattern similar to those of currents triggered by the specific agonist GSK1016790A, and inhibition by the TRPV4 antagonist HC067047. We found that TRPV4 responses to LPS were significantly smaller and less

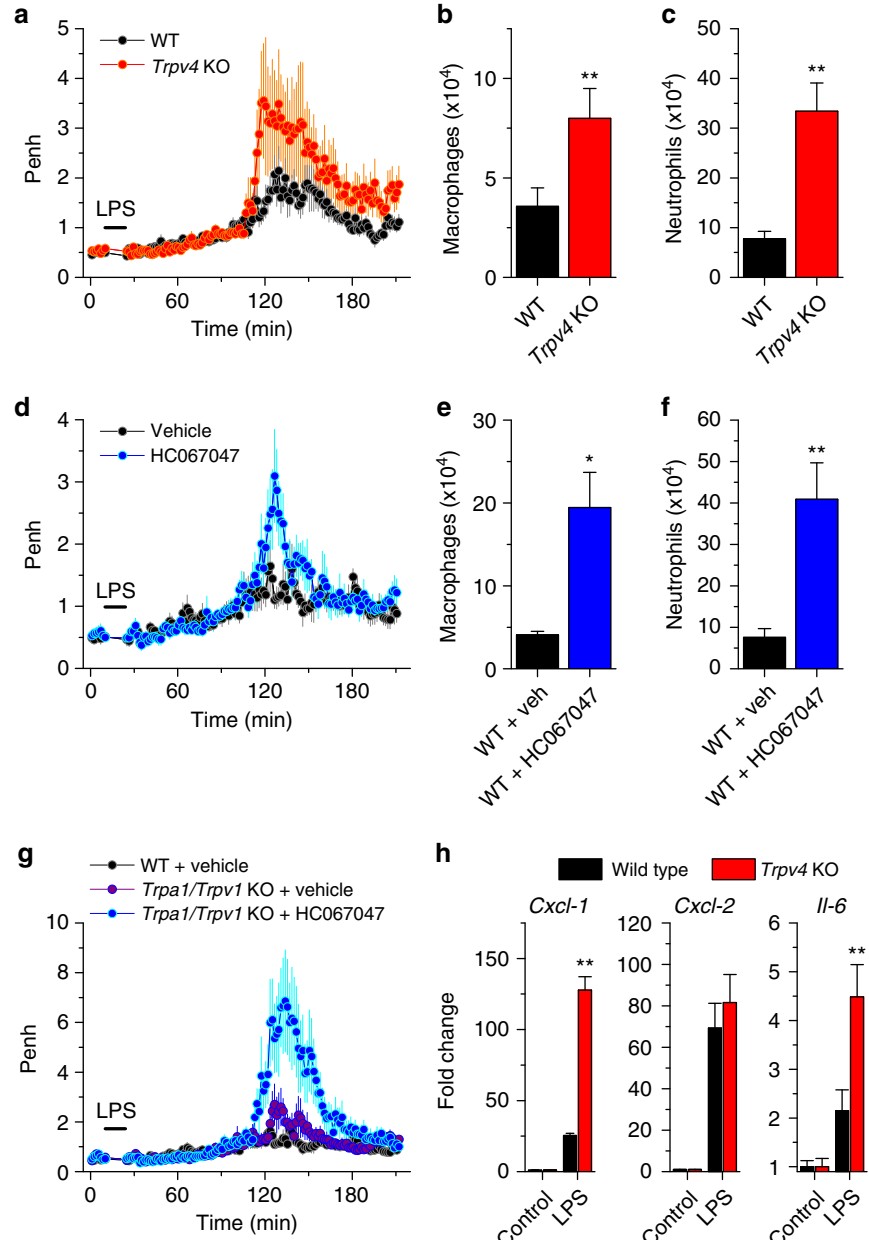

**Fig. 7** TRPV4-deficient mice display exacerbated airway responses to LPS. **a** Time course of average enhanced pause (Penh) determined in unrestrained whole-body plethysmography experiments performed in wild type (WT, $n = 10$) and $Trpv4$ KO ($n = 7$) mice exposed to aerosols of LPS. The area under the curve (AUC) of $Trpv4$ KO traces is significantly larger than AUC from WT ($144 \pm 56 > 47 \pm 14$; $P = 0.048$, unpaired $t$-test). **b, c** Average number of macrophages (**b**) and neutrophils (**c**) in the bronchoalveolar lavage fluid collected 3 h after LPS exposure. **$P < 0.01$, two-tailed $t$-test. **d** Average Penh determined in unrestrained whole-body plethysmography experiments performed in WT mice pretreated with vehicle (0.8% DMSO, $n = 7$) or the TRPV4 inhibitor HC067047 (10 mg kg$^{-1}$ in 0.8% DMSO, $n = 8$). The vehicle or HC067047 were administered intraperitoneally 15 min previous to the LPS challenge. AUC$_{Vehicle} = 23 \pm 4 <$ AUC$_{HC067047} = 54 \pm 14$; $P = 0.039$, unpaired $t$-test. **e, f** Average number of macrophages (**e**) and neutrophils (**f**) in the bronchoalveolar lavage fluid collected 3 h after LPS exposure. $Veh$, vehicle. *$P < 0.05$; **$P < 0.01$, two-tailed $t$-test. **g** Average Penh of WT and $Trpa1/Trpv1$ KO mice pretreated with vehicle (0.8% DMSO, $n = 6$) or the TRPV4 inhibitor HC067047 (10 mg kg$^{-1}$ in 0.8% DMSO, $n = 6$). The vehicle or HC067047 were administered intraperitoneally 15 min previous to the LPS challenge. AUC$_{(WT)\ Vehicle} = 23 \pm 4 <$ AUC$_{(Trpa1/Trpv1\ dKO),\ Vehicle} = 88 \pm 16$; $P = 0.0018$, unpaired $t$-test. AUC$_{(Trpa1/Trpv1\ dKO),\ Vehicle} <$ AUC$_{(Trpa1/Trpv1\ dKO),\ HC067047} = 257 \pm 65$; $P = 0.031$, unpaired $t$-test. **h** Fold change of mRNA transcripts of $Cxcl-1$, $Cxcl-2$, and $Il-6$ in mTEC untreated (Control) or treated during 3 h with LPS (20 µg ml$^{-1}$). Cytokine cycle thresholds were normalized to β-actin signals. Fold change is relative to untreated cells. **$P < 0.01$, two-tailed $t$-test. In all experiments mice were challenged with aerosolized LPS (50 µg ml$^{-1}$) during 15 min

prevalent than those to GSK1016790A, which is consistent with the fact that the former compound is a weaker channel agonist[29]. However, the potency of LPS on TRPV4 ($11\,\mu g\,ml^{-1}$) is within the same range of that previously reported for TRPA1 ($\sim 3\,\mu g\,ml^{-1}$)[8]. Also similar to what we found for TRPA1[8], the free availability of the lipidic region of LPS is required for TRPV4 activation. However, in contrast to TRPA1, TRPV4 is activated by lamellar-shaped LPS from *S. minnesota*. Although the precise molecular mechanism underlying the activation of these channels by LPS remains elusive, we may speculate that TRPV4 is more sensitive than TRPA1 to mechanical perturbations induced by LPS in the plasma membrane. Importantly, the concentrations at which LPS activates TRPV4 (above $3\,\mu g\,ml^{-1}$) are compatible with those used in experimental models of LPS-induced airway inflammation (e.g., $250\,\mu g\,ml^{-1}$[30] and $500\,\mu g\,ml^{-1}$[31] and very recently in a study on the protective role of LPS against allergy to house dust mite, ranging $2–20\,\mu g\,ml^{-1}$[16]). Thus, our findings are relevant to current experimental models of endotoxin challenge.

It has been shown that intestinal EC lines secrete cytokines and chemokines several hours after stimulation with the synthetic TRPV4 agonist 4αPDD[32], and that mouse macrophages release NO 30 min after 4αPDD application[33]. However, the immediate cellular consequences of TRPV4 stimulation have received less attention and the effects of natural agonists of this channel remain unknown. Given that TRPV4 activation occurs within seconds of exposure to LPS, we here focused on the acute phase of the cellular responses, particularly within the first few minutes of LPS challenge. We found that LPS triggers TRPV4-dependent NO production through the activation of NOS, namely nNOS and iNOS. Both isoforms have been previously reported to be expressed in airway ECs[34–36], although the direct $Ca^{2+}$ dependence of nNOS[19] suggests this isoform as the major contributor to the NO production upon TRPV4-induced $Ca^{2+}$ increase. Interestingly, nNOS has been recently localized in the proximal part of the ciliary axoneme[36], where TRPV4 expression plays a role in controlling the beating frequency. We found increased beating frequency in ciliated mTEC upon LPS challenge. However, this effect was not dependent of NOS-derived NO. On the other hand, NO produced by TRPV4 stimulation with LPS induced a direct anti-bacterial effect, as expected from previous reports[37]. Furthermore, NO induces a number of additional protective responses[38], including those we show here that depend of TRPV4, i.e., regulation of neutrophil infiltration (see below) and bronchodilation.

LPS-induced TRPV4-dependent increase in CBF was comparable to that triggered by other stimuli of TRPV4, including

### Table 1 List of TaqMan probes

| Gene name | Probe ID (in Applied Biosytems) |
| --- | --- |
| Trpc6 | Mm01176083_m1 |
| Trpa1 | Mm00625268_m1 |
| Trpv1 | Mm01246302_m1 |
| Trpv2 | Mm00449223_m1 |
| Trpv3 | Mm00454996_m1 |
| Trpv4 | Mm00499025_m1 |
| Trpv5 | Mm01166037_m1 |
| Trpv6 | Mm00499069_m1 |
| Trpm3 | Mm00616485_m1 |
| Trpm7 | Mm00457998_m1 |
| Trpm8 | Mm00454566_m1 |
| Il-6 | Mm00446190_m1 |
| Cxcl-1 | Mm04207460_m1 |
| Cxcl-2 | Mm00436450_m1 |
| β-actin | 4352341E |

heating and the synthetic chemical agonists 4αPDD[24] and GSK1016790A[39]. However, our results unveiled the first naturally occurring chemical agonist of TRPV4 inducing an increase in ciliary movement and suggests that TRPV4 activation by LPS enhances the clearance of pathogens from the airways.

In parallel with nNOS, cytosolic iNOS activity is upregulated in response to LPS-induced TRPV4 activation. Our confocal microscopy images revealed the presence of iNOS in aggresome-like structures in human airway epithelial 16HBE cells, similar to those described in rat aortic vascular smooth muscle cells[21]. Thus, our data suggests that the increase in intracellular $Ca^{2+}$ concentration that results from LPS-induced activation of TRPV4 stimulates the release of active iNOS from inactive aggresomes through the activation of $Ca^{2+}$-calmodulin kinase II $\delta_2$ ($CaMKII\delta_2$)[21]. The differential compartmentalization of NOS isoforms might indicate that NOS-derived NO pools may operate distinctively. A direct activation of nNOS by the TRPV4-mediated increase in intracellular $Ca^{2+}$ concentration can quickly induce increase in luminal NO levels and increase CBF. On the other hand, TRPV4-triggered iNOS disaggregation and consequent activation can reinforce NO diffusion to the lumen of the airways, but can also contribute to NO diffusion to the basal layer of the ECs.

Acute inhalation of LPS activates the TLR4 pathway, leading to changes in the ventilatory pattern and to a severe inflammatory response characterized by activation of alveolar macrophages and recruitment of polymorphonuclear leukocytes into the airways[40–43]. We here present genetic and pharmacological evidence demonstrating that TRPV4 activation limits the magnitude of these responses. This key role of TRPV4 is compatible with its high expression in airway ECs, which are readily accessible to luminal LPS. Our data also suggest that TRPA1 and TRPV1, two main receptors of noxious chemicals in the airways[44–46], may contribute to the LPS-induced ventilatory changes, although to a lesser extent.

The mechanisms underlying the limiting role of TRPV4 in ventilatory changes and leukocyte infiltration into the airways induced by LPS may be based partly on the TRPV4-dependent production of NO, discussed above. NO, most likely the pool produced by iNOS, is known to induce bronchodilation through its relaxing action on airway smooth muscle[47, 48] and to decreased leukocyte adhesion on endothelial cells[49–53] through down-regulation of cell adhesion molecules[54–56]. The protective role of the LPS-TRPV4-iNOS mechanism we unveiled here is in line with the exacerbated inflammatory responses to LPS observed in *Inos*-null mice[31]. Together with the absence of NO, increased chemotractant levels observed in *Trpv4* KO mTEC might be the responsible for the exacerbated neutrophil infiltration upon LPS challenge.

It has been proposed that inhibition of TRPV4 might be beneficial for the treatment of multiple respiratory conditions, including chronic heart failure, hypoxia-induced pulmonary hypertension, acute lung injury, chronic obstructive pulmonary disease, and cough[45, 57, 58]. However, our results demonstrate that inhibition of this channel abrogates protective responses in mouse airway ECs and exacerbates mouse ventilatory and inflammatory responses to LPS challenge. Furthermore, we found that the TRPV4-mediated intracellular $Ca^{2+}$ increase and subsequent NO production in response to LPS operates also in freshly isolated human nasal ECs. This argues for the need of therapies excluding the inhibition of this channel in airway ECs, especially in conditions that may be associated with respiratory infections with gram-negative bacteria.

In summary, our work shows that LPS activates TRPV4 and that this triggers immediate protective responses in airway ECs, unveiling this channel as a player in innate defense responses.

Our findings demonstrate that, not only sensory neurons, but also non-excitable cells are endowed with TRP channel-mediated mechanisms of detection of major bacterial cues. These mechanisms trigger very fast, cell-intrinsic events, distinct from the currently established ones based on cytokine secretion regulating the homing and activation of immune cells.

## Methods

**Transfections**. For intracellular $Ca^{2+}$ imaging experiments, HEK293T (from American Type Culture Collection) cells were transiently transfected with the mouse TRPV4, in the bicistronic pCAGGS/IRES-GFP vector, using the Mirus TransIT-293 kit (Sopachem n.v./s.a, Eke, Belgium).

**Intracellular $Ca^{2+}$ and nitric oxide imaging experiments**. Cells were incubated with Fura-2 acetoxymethyl ester for 30 min at 37 °C. For recordings, bath solutions prepared in Krebs (containing (in mM): 150 NaCl, 6 KCl, 1.5 CaCl₂, 1 MgCl₂, and 10 HEPES, 10 glucose and titrated to 7.4 with NaOH) were perfused by gravity via a multi—barreled pipette tip. Intracellular $Ca^{2+}$ concentration was monitored through the ratio of fluorescence measured upon alternating illumination at 340 and 380 nm using an MT-10 illumination system and the Xcellence pro software (Olympus Belgium N.V., Berchem, Belgium). The TRPV4 inhibitor HC067047 (10 μM) was perfused prior to LPS applications, as indicated in the Figures. The TLR4 inhibitor Cli-095 (1 μM) was added to the cells 20 min before measurements, and constant perfusion of this compound was kept during the measurements. PMB (300 μg ml⁻¹) was added to LPS solutions and mixed for 30 min.

For simultaneous $Ca^{2+}$ and NO imaging, cells were incubated with Fura-2 acetoxymethyl ester as detailed above and were also pre-loaded with 4-Amino-5-Methylamino-2',7'-Difluorofluorescein (DAF-FM) diacetate for 20 min. DAF-FM images were acquired using the excitation/emission wavelengths at ~488/520 nm. The fluorescence intensity of individual cells in each experiment is expressed as percentage of the average signal recorded at baseline ($F/F_0$, %).

**Patch-clamp**. The whole-cell patch-clamp recordings were made using an EPC-7 patch-clamp amplifier (HEKA, Lambrecht/Pfalz, Germany) and the Clampex 9.0 software program (Axon instruments, Sunnyvale, CA, USA). Currents were filtered at 5 kHz, acquired at 10 kHz, and stored for off-line analysis on a personal computer. In order to minimize voltage errors, the series resistance was compensated by 30–50% and the capacitance artifact was reduced using the amplifier circuitry. Membrane TRPV4 currents were elicited by a 400 ms-long voltage ramp from −125 mV to + 100 mV every 5 s. Before recordings, cells were allowed to stabilize in Krebs solution. During recordings cells were perfused with extracellular solution (in mM): 140 NaCl; 5 CsCl; 2 CaCl₂; 1 MgCl₂; 10 HEPES, 10 glucose monohydrate and pH adjusted to 7.4 with NaOH. The pipette solution contained (in mM): 110 cesium methane sulfonate; 25 CsCl; 30 HEPES; 0.362 CaCl₂; 2 MgCl₂ and pH adjusted to 7.2 with CsOH.

**$Ca^{2+}$ imaging and ciliary beat frequency in mTEC**. Tracheobronchial ECs (containing ciliated cells) were isolated from WT and *Trpv4* KO mouse trachea and cultured following the protocol described elsewhere[24, 59]. Briefly, trachea are dissected, opened lengthwise and placed onto 0.15% pronase solution for overnight digestion. Next day, cells are collected by gentle rocking of the pronase solution, and transferred to 20% FBS Ham-F12 medium. After centrifugation at 1400 rpm, 10 min at 4 °C, cells are rinsed in DMEM/F12 supplemented with penicillin/streptomycin, HEPES (15 mM), glutamine (4 mM), and 0.0375% NaHCO₃ (mTEC Basic medium) and cultured for 5 h in Primaria Plates (Thermo Fisher Scientific, Rockford, IL, USA). Dishes are gently rinsed with mTEC Basic medium and cells are collected and centrifuged at 1400 rpm, 10 min at 4 °C. After centrifugation, cells are resuspended in 5% FBS mTEC Basic Medium supplemented with insulin (20 μg ml⁻¹), epidermal growth factor (40 ng ml⁻¹), bovine pituitary extract (60 μg ml⁻¹), transferrin (5 μg ml⁻¹), cholera toxin (100 ng ml⁻¹), and retinoic acid (0.1 μM), and seeded in new culture plates. ECs are used after 2 days in culture.

For $Ca^{2+}$ imaging experiments and NO measurements, isolated cells were seeded in glass coverslips pre-coated with collagen I. CBF was measured in primary cultures ciliated cells using with a high-speed digital imaging system as previously described[24]. Briefly, phase-contrast images (512 × 512 pixels) were collected at 120–135 frames per second with a high-speed CCD camera using a frame grabber (Infaimon, Barcelona, Spain) and recording software from Video Savant (IO Industries, London, ON, Canada). The CBF was determined from the frequency of variation in light intensity of the image as a result of repetitive motion of cilia.

**Bacterial live-dead assay**. Bacterial live-dead assay was adapted from a protocol described by Lee and colleagues[11]. Overnight cultures of *Escherichia coli* (DH5α) were diluted in LB medium (in g L⁻¹: 10 Tryptone; 5 yeast extract; 10 NaCl) and grown to log phase (OD = 0.1, ~1 h). Bacteria (1 ml) were resuspended in low-salt (50%) saline supplemented with 0.5 mM glucose, 1 mM HEPES and 1.5 mM CaCl₂ (pH 7.4) and incubated with shaking for 60 min. Bacteria in 200 μl of this solution were added to mTEC cells previously grown to confluence in 12-well plates. Where

indicated, epithelial cultures were pre-incubated for 15 min with the TRPV4 inhibitor HC067047 (1 μM) or the NOS inhibitor L-NAME (1 mM). Bacteria were allowed to settle for 10 min and 100 μl of solution were carefully removed. After incubation for 2 h at 37 °C, bacteria were collected in 1 ml of supplemented saline and centrifuged at 5000 rpm for 5 min. Bacteria pellets were later resuspended in 30 μl of a 1:1 solution of SYTO 9 and propidium iodide (PI) (LIVE/DEAD *Bac*-Light Bacterial Viability Kit; Invitrogen, Merelbeke, Belgium). This 1:1 mixed solution allowed identifying dead cells as red, whereas all cells (living and dead cells) were stained in green. Cells were mounted in glass slides and visualized using an Olympus microscope. At least 5 randomly selected fields per condition were imaged. Stainings were quantified by determining the area of positive red and green fluorescence for each image using a custom-made routine in Matlab (The Math-Works, Inc., MA, USA). Since SYTO9 stains both living and dead bacteria, we calculated the percentage of dead cells as Area_PI/Area_SYTO9. Control experiments for dead bacteria were performed by incubating 0.1 OD bacterial culture in isopropanol for 30 min. Simultaneously, 0.1 OD bacteria were kept in saline solution to assess spontaneous cell death.

**Western blot**. Freshly isolated splenocytes, freshly isolated mouse airway ECs and 16HBE cells (kindly provided by Dr. Gruenert, University of California, San Francisco, USA) were lysed in ice for 30 min using fresh prepared lysis buffer containing: 150 mM NaCl, 1.0% Triton X-100, 0.5% sodium deoxycholate, 0.1% SDS, 50 mM Tris, pH 8.0, supplemented with protease inhibitor cocktail (Westburg BV, Leusden, The Netherlands). Proteins were collected from the supernatant after centrifugation and protein concentration was determined with the bicinchoninic acid protein assay kit (Thermo Scientific, Waltham, MA, USA). Equal protein amounts (20 μg) were resolved on precast NuPAGE gels (Invitrogen) and transferred to polyvinylidine difluoride membranes. Membranes were further probed with iNOS (1:1000; #2977), nNOS (1:1000; #4234), eNOS (1:1000; clone 49G3, #9586) and β-actin (1:10,000; clone AC-15, Sigma-Aldrich) specific antibodies. Anti-mouse (1:10,000; #7076) or anti-rabbit (1:10,000; #7074) secondary antibodies were incubated for 1 h at room temperature. Signals were visualized by enhanced chemiluminescence according to the manufacturers' instructions (ECL kit, Invitrogen). Antibodies against NOS isoforms and secondary antibodies were purchased from Cell Signaling Technology (Leiden, Netherlands). Original scans of probed membranes are shown in Supplementary Fig. 15.

**Confocal microscopy**. 16HBE cells were seeded in glass coverslips and exposed to LPS or to the TRPV4 agonist GSK1016790A during 5 min. When indicated, cells were pre-incubated for 15 min with the TRPV4 inhibitor HC067047 (10 μM). After treatment, cells were fixed with cold paraformaldehyde and permeabilized with 0.2% Triton X-100. Primary antibody against iNOS (1:1000; PA3-030A, Thermo-Fisher Scientific) was incubated overnight at 4 °C, followed by anti-rabbit Alexa Fluor Plus 488 (1:1000; A32731, Invitrogen) for 1 h at room temperature. Coverslips were mounted in glass slides using DAPI-containing mounting solution (VectaShield, Vector Laboratories, Burlingame, CA, USA). The confocal images of labeled cells were collected using the optimal pinhole size for the ×63 oil objective of a Zeiss LSM 510 Meta Multiphoton microscope (Carl Zeiss AG, Oberkochen, Germany). We used a custom-designed software in Matlab (The MathWorks, Inc) to determine the number and size of iNOS aggresomes.

**Isolation and culture of human nasal epithelial cells**. Nasal ECs (NEC) were isolated from the inferior turbinates of control subjects as previously reported[60]. Control subjects were non-smokers, asymptomatic for allergic rhinitis or rhinosinusitis. Detailed subject information can be found in Steelant et al. (2016)[61]. Briefly, tissue was washed in sterile saline and enzymatically digested in 0.1% Pronase (Protease XIV, Sigma) solution in DMEM/F12 medium containing 100 U mL⁻¹ penicillin, 100 μg ml⁻¹ streptomycin, and 2% Ultroser G (Pall Life Sciences, Zaventem, Belgium). After overnight digestion at 4 °C, cells were washed and centrifuged at 100×g for 5 min. Pelleted cells were resuspended in 10 ml of culture medium and incubated in a plastic culture flask for 1 h at 37 °C to remove fibroblasts. NECs in suspension were further purified by negative selection with CD45 and CD15 magnetic beads (Dynabeads; Invitrogen), according to the manufacturer's instructions. NECs (2 × 10⁵) were plated in 50 μg ml⁻¹ collagen-coated glass coverslips and used for experiments after 48 h in culture.

**Whole-body plethysmography**. The ventilatory function of mice was monitored using unrestrained whole-body plethysmography (EMKA Technologies, Paris, France). LPS was delivered using a PARI BOY nebulizer (PARI GmbH, Starnberg, Germany) for 15 min. After recording, mice were deeply anesthetized by intraperitoneal injection of pentobarbital and the lungs were washed three times with sterile 0.9% NaCl solution. Cells recovered from lung lavages were counted and spun onto microscope slides, air-dried, and stained. For each sample we determined the number of macrophages, eosinophils, neutrophils, and lymphocytes. To test the effects of the TRPV4 inhibitor HC067047 this compound was injected intra peritoneal (vehicle HBSS + 0.8% DMSO) 15 min before the experiments. Lung lavages and cell counting were performed by different researchers, both blinded to mouse genotypes.

**RT-PCR**. Total RNA from cultured mTEC was extracted using the RNeasy Mini Kit (Qiagen, Antwerp, Belgium), following manufacturer's protocol. RNA concentration was assessed using a NanoDrop ND 1000 Spectrophotometer (Nano-Drop Technologies, Wilmington, DE, US). cDNA synthesis was performed with 1 μg of total RNA using the Ready-To-Go You-Prime First-Strand Beads (GE Healthcare, Diegem, Belgium). Quantitative PCR reactions (20 μl), containing 3 μl cDNA template (diluted 1:5), Universal TaqMan MasterMix (2x concentrated, Life Technologies), specific TaqMan probes (Table 1, 20x concentrated, Life Technologies) and $H_2O$, were performed with the 7500 Fast Real-Time PCR System (Life Technologies). Reactions were made using the following program: 50 °C for 2 min and 95 °C for 10 min, followed by 40 cycles of 95 °C for 15 sec and 60 °C for 1 min. Non-template controls (NTCs) were used as negative controls in every experiment.

For cytokine expression experiment, mTEC from WT and *Trpv4* KO mice were incubated with LPS (20 μg ml$^{-1}$) during 3 h prior to RNA isolation protocol.

**Animals**. Ten- to 12-week-old male mice were used in all experiments. WT C57Bl/6 mice were obtained from Janvier (France). *Trpv4* KO mice and *Tlr4* KO mice were kindly provided by Prof. W. Liedtke (Duke University, Durham, NC, USA) and by Profs. B. Lambrecht and H. Hammad (VIB, Ghent, Belgium). *Trpa1/Trpv1* double KO mice were obtained from an in-house breeding program. All knockout mice were backcrossed at least 10 times in the C57Bl/6 background. Mice were housed under identical conditions, with a maximum of four animals per cage on a 12-h light–dark cycle and with food and water *ad libitum*. All animal experiments were carried out in accordance with the European Union Community Council guidelines and were approved by the local ethics committees.

**Reagents**. All chemicals were purchased from Sigma-Aldrich (Bornem, Belgium). We used LPS extracted from *E. coli*, strain 0127:B8.

**Statistics**. Magnitudes were statistically compared using GraphPad Prism v.5 (GraphPad Software) as indicated in Figure legends. Differences were considered to be statistically significant when $P < 0.05$. Normality was not checked prior parametric tests. However, when parametric test were used, non-parametric tests were carried out in parallel on data sets yielding the same conclusions. No statistical method was used to predetermine sample sizes. Nonetheless, our sample sizes are similar to those used in generally used in the field. The variance was generally similar among compared groups. No blinding/randomization was performed.

**Data availability**. All relevant data from this article are available from the authors upon reasonable request.

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

## Acknowledgements

We would like to thank Melissa Benoit, Silvia Pinto, and Evy Van Den Broek for the maintenance of the cell cultures and mouse colonies. Y.A.A. held a Postdoctoral Mandate of the KU Leuven and is currently a Postdoctoral Fellow of the Fund for Scientific Research Flanders (FWO). B.B. was funded by a Ph.D. grant of the Agency for Innovation by Science and Technology (IWT). Research was supported by grants from the Belgian Federal Government (IUAP P7/13), the FWO (G.0702.12, 1.5.068.16 N) and the Research Council of the KU Leuven (Grants GOA/14/011 and PF-TRPLe), the Spanish Ministry of Economy and Competitiveness (SAF2015-69762-R and María de Maeztu Programme for Units of Excellence in R&D MDM-2014-0370), Fondo de Investigación Sanitaria (RD12/0042/0014), and the FEDER Funds.

## Author contributions

Y.A.A. and K.T. conceived and designed the project. Y.A.A. performed NO quantification, western blotting, and confocal imaging. Y.A.A., R.N. and A.S. conducted Ca$^{2+}$ imaging experiments. B.B., V.M.M., J.L.A. and K.T. designed or conducted electrophysiological recordings. C.J., C.P. and M.A.V. designed and performed Ca$^{2+}$ imaging experiments and CBF measurements. Y.A.A. and A.L.-R. conducted the bacteria dead/live assay. Y.A.A., K.L. and P.H.M.H. conducted experiments in the 16HBE cell line. B.S. and P.W.H. isolated and cultured primary human NECs. Y.A.A., V.D.V. and J.A.J.V. conceived or performed in vivo experiments and leukocyte quantification. Y.A.A. and K.T. wrote the manuscript. B.N. and T.V. contributed to the interpretation of data. All authors edited the manuscript.
