## [Peer Review file · Nature Communications]

Reviewers' comments:

Reviewer #1 (Remarks to the Author):

TRPV4 activation triggers protective responses to bacterial LPS in airway epithelial cells

Overall a well constructed manuscript examining the activation of TRPV4 by LPS. Utilizing mouse nasal epithelial cells from WT and TRPV4 KO mice the investigators demonstrate that TRPV4 is rapidly activated by LPS yielding a rise in intracellular calcium with concomitant production of intracellular nitric oxide and an increase in ciliary beat frequency.

Introduction:

Line 37/38 does not read correctly

Line 47 Gram positive is incorrect, should be gram-positive. Same with Gram negative ,should be gram-negative

Line 53/54 is incorrect -- see PMID 20133764, 23041624

Results:

Figure 3A, 3C, and 3D – it would be helpful to have a legend directly on the figure showing what each color line is representing.

Discussion – TRPV4 activation yields NO and they discuss that it is bactericidal. They should demonstrate the amount of NO stimulated by TRPV4 activation is actually bacteriocidal (PMID 23041624)

The authors mention that TRPV4 is highly expressed in the respiratory epithelium. Why do they think only 50% of cells respond to TRPV agonists?

Is the rise in ciliary beat frequency driven by the rise in intracellular calcium or NO? Both second messengers are known to increase CBF. Fairly easy experiment to identify which pathway is driving increase in CBF.

Does any of this happen in human airway cells which would make it potentially clinically relevant.

Methods: Identify mouse breeding lines in "animals" section of the methods (373)

Reviewer #2 (Remarks to the Author):

A perfectly smooth paper, state-of-the-art methods, consistent results, appropriate controls, convincing conclusions, and relevant news about an immediate defense strategy of respiratory epithelial cells mediated by TRPV4, faster acting than the innate immunity typified by the toll-like receptors and even faster than the „toll-likely“ unselective TRPA1 receptor of nociceptive tracheal nerve endings evoking `neurogenic inflammation´ when challenged by LPS from gram-negative bacteria.

I'm only missing two little considerations:

In their previous paper on LPS activating TRPA1, the authors provided some evidence that the lipid A moiety may integrate in the plasma membrane raft and exert force from lipid` on the ion channel, activating or, at least, sensitizing it. Do they have any similar evidence in case of TRPV4 that also appears sensitive to membrane stretch?

Impairment of the epithelial defense by TRPV4 deletion or block enhanced the ventilatory disturbance evoked by LPS aerosol challenge in a way that indicates bronchoconstriction. Do the authors have any evidence that neuropeptides (CGRP or SP) released by TRPA1 activation are involved in this reaction?

Reviewer #3 (Remarks to the Author):

Comments:

The intent of this manuscript entitled "TRPV4 activation triggers protective responses to bacterial lipopolysaccharides in airway epithelial cells" was to determine the protective role of TRPV4 in LPS-induced inflammatory innate immune reaction. LPS application led to immediate protective responses in airway epithelial cells, including direct antimicrobial action, increase in airway clearance, and the regulation of the inflammatory innate immune reaction, which are dependent on TRPV4-mediated rapid intracellular Ca²⁺ increase. The proper inhibitive and gene knockout data support the hypothesis and conclusion in this study. This is an interesting paper with the potential to have a significant impact on the field. However, I have some concerns related to the experimental design and data organization in this study that may need authors' attention to improve the manuscript.

Major concern:

1. The biggest concern of this study is that the figures are not well organized which gives the manuscript a messy appearance. There are too many supplemental figures, most of which can be reorganized into a single figure by optimizing the size of the panels. For example, Sfigure 2b and Sfigure 3 should be combined with figure 1 as they are corresponding traces to the bar graphs in figure 1b and c. The authors should also provide Ca²⁺ responding traces to match the "WT+HC" bar in figure 1b. Sfigure 6 should be merged with figure 3 to match figure 3c and d. In figure 4, the authors should provide corresponding Ca²⁺ responding traces and NO responding traces to match panel c and d. Sfigure 8 should be included in figure 5 to match figure 5e. In addition, the authors should provide Ca²⁺ responding traces and NO responding traces to match e in figure 5.
2. Since protein expression level of TRPM7 is relatively high compared with other TRP channels in both mTEC and 16HBE, why didn't the authors perform any experiments to rule out the possibility of TRPM7 involvement in LPS-induced intracellular free Ca²⁺ increase? This study should provide evidence for this concern using TRPM7 activator, inhibitor or overexpression of TRPM7 in HEK293T as shown in Sfigure 9c in which TRPV3 was excluded from being activated by LPS.
3. The experiments of TRPV4-transfected HEK293T used to confirm the role of mouse TRPV4 in LPS-induced intracellular free Ca²⁺ increase in mouse TEC should be included with the supplemental materials.
4. In this study, Tlr4 pathway was excluded from the activation of TRPV4 by LPS. The authors provides sufficient evidence, however, the data organization is messy. The middle pane in figure 1a, figure 1b and Sfigure 6 should be moved to figure 3 together with Tlr4 inhibitive data, which would make the figure or evidence appear uniform and integrated.
5. The third section of results mentioned that the LPS-induced increase in ciliary beat frequency is TRPV4 dependent. Can the author also include supplemental video data to support the summarized data shown in Sfigure 7?
6. In figure 5f, besides using TRPV4 inhibitor HC067047 to inhibit iNOS disaggregation, the authors should also use TRPV4 knockdown or knockout cells to further confirm the conclusion.

Minor concerns:

1. In figure 1b, please show standard deviation (SD) or standard error (SE) for each bar.
2. The size of panels in each figure should be optimized to make the figures neat.

TRPV4 activation triggers protective responses to bacterial LPS in airway epithelial cells. Alpizar Y. A. et al.

Answer to the Reviewers

We are very grateful to all reviewers for their critical evaluation of our work. We have found all comments and suggestions enormously valuable. The answers to every point are included below. The changes introduced in the manuscript are indicated in red font.

Reviewer #1:

1. Line 37/38 does not read correctly.

These lines have been changed to: "Covered with a mucociliary layer and connected by tight junction proteins, EC serve as a structural barrier against inhaled pathogens, and control the screening of the luminal microenvironment by antigen-presenting cells¹"

2. Line 47 Gram positive is incorrect, should be gram-positive. Same with Gram negative, should be gram-negative.

Corrected.

3. Line 53/54 is incorrect -- see PMID 20133764, 23041624.

The Introduction has been re-written to include these two papers.

4. Figure 3A, 3C, and 3D – it would be helpful to have a legend directly on the figure showing what each colour line is representing.

These legends have been added.

5. TRPV4 activation yields NO and they discuss that it is bactericidal. They should demonstrate the amount of NO stimulated by TRPV4 activation is actually bactericidal (PMID 23041624).

Following the experimental procedure suggested by the reviewer, we now provide direct evidence that NO produced by activation of TRPV4 in epithelial cells in contact with bacterial cultures exerts a bactericidal effect. We show that 0.1 OD cultures of *E. coli* exhibit 12% of dead cells when incubated with a monolayer of mouse tracheobronchial epithelial cells. This effect was abrogated when the co-culture was done in the presence of a TRPV4 or NOS inhibitors (included in new Figure 6). Thus, we demonstrate that both TRPV4 and NOS are required for the anti-bacterial effect.

On a technical note, we would like to mention that we have modified the experimental settings originally described by Lee *et al.*, 2012. First, following instructions of the manufacturer, we have adjusted the ratios of SYTO9 and PI to stain the whole bacterial population in green (SYTO9) and the dead population in red (PI). Second, we have used *E. coli* instead of *Pseudomonas aeruginosa* to be consistent with all other results. We noticed that we obtained lower proportions of dead cells than those reported by Lee *et al.* Although it was not an objective to identify the reasons behind this difference, we think that working in aerobic conditions so as to resemble the scenario of the airways might have reduced the lifespan of reactive NO in our settings.

6. The authors mention that TRPV4 is highly expressed in the respiratory epithelium. Why do they think only 50% of cells respond to TRPV agonists?

We have been confronted with this apparent paradox in many opportunities. A few years ago we decided to investigate this issue and found that indeed, in Ca^{2+} imaging experiments, relatively weak TRP channel agonists (as is the case for LPS on TRPV4, if compared to GSK1016790A) do not necessarily stimulate all cells expressing these channels. This study was published under the title “*Lack of correlation between the amplitudes of TRP channel-mediated responses to weak and strong stimuli in intracellular Ca^{2+} imaging experiments*” (Alpizar *et al.*, *Cell Calcium* 2013, 54(5):362-74).

The abstract of this paper reads as follows:

“It is often observed in intracellular Ca^{2+} imaging experiments that the amplitudes of the Ca^{2+} signals elicited by newly characterized TRP agonists do not correlate with the amplitudes of the responses evoked subsequently by a specific potent agonist. We investigated this rather controversial phenomenon by first testing whether it is inherent to the comparison of the effects of weak and strong stimuli. Using five well-characterized TRP channel agonists in commonly used heterologous expression systems we found that the correlation between the amplitudes of the Ca^{2+} signals triggered by two sequentially applied stimuli is only high when both stimuli are strong. Using mathematical simulations of intracellular Ca^{2+} dynamics we illustrate that the innate heterogeneity in expression and functional properties of Ca^{2+} extrusion (e.g. plasma membrane Ca^{2+} ATPase) and influx (TRP channels) pathways across a cellular population is a sufficient condition for low correlation between the amplitude of Ca^{2+} signals elicited by weak and strong stimuli. Taken together, our data demonstrate that this phenomenon is an expected outcome of intracellular Ca^{2+} imaging experiments that cannot be taken as evidence for lack of specificity of low-efficacy stimuli, or as an indicator of the need of other cellular components for channel stimulation.”

We have summarised this observation during the discussion by stating that (lines 247-249):
“*We found that TRPV4 responses to LPS were significantly smaller and less prevalent than*

those to GSK1016790A, which is consistent with the fact that the former compound is a weaker channel agonist (Alpizar et al. 2013).'

7. Is the rise in ciliary beat frequency driven by the rise in intracellular calcium or NO? Both second messengers are known to increase CBF. Fairly easy experiment to identify which pathway is driving increase in CBF.

In new measurements of CBF in mTEC pre-incubated with NOS inhibitor L-NAME we found that NO is dispensable for the increase in CBF (Supplementary Fig. 10).

8. Does any of this happen in human airway cells, which would make it potentially clinically relevant?

The first version of the manuscript included results in the human bronchial epithelial cell line 16HBE. We have now assessed the effect of LPS in primary cultured human nasal epithelial cells (NEC) isolated from inferior turbinate of healthy, non-smoker individuals. We found that, as observed in mTEC and 16HBE, these cells are stimulated by LPS and the increase in intracellular Ca^{2+} is followed by production of NO. These results are included in Supplementary Fig. 4.

10. Identify mouse-breeding lines in 'animals' section of the methods (373)

The backgrounds of mice are now specified in this section (Line 455).

Reviewer #2 (Remarks to the Author):

1. In their previous paper on LPS activating TRPA1, the authors provided some evidence that the lipid A moiety may integrate in the plasma membrane raft and exert 'force from lipid' on the ion channel, activating or, at least, sensitizing it. Do they have any similar evidence in case of TRPV4 that also appears sensitive to membrane stretch?

In our previous work (Meseguer et al. 2014) we showed that TRPA1 was most sensitive to conically-shaped LPS (i.e.: *E. coli*), whereas lamellar LPS extracted from *S. minnesota* is less potent in activating the channel. In the revised version of this manuscript we now report that TRPV4 is similarly sensitive to *E. coli* and *S. minnesota* LPS (included in Figure 3e,f). Although the nature of the interaction LPS-membrane-TRP remains unknown, the better-documented mechano-sensitivity of TRPV4 could explain its increased sensitivity to LPS when compared to TRPA1. The elucidation of the molecular mechanism underlying TRPV4 (and TRPA1) activation will require experiments extending beyond the scope of our current study.

2. Impairment of the epithelial defence by TRPV4 deletion or block enhanced the ventilatory disturbance evoked by LPS aerosol challenge in a way that indicates bronchoconstriction. Do the authors have any evidence that neuropeptides (CGRP or SP) released by TRPA1 activation are involved in this reaction?

This is certainly an interesting question, although we believe that a complete study on the role of TRPA1 in LPS-induced airway responses is outside the scope of the present communication. Nevertheless, we explored the possibility of both TRPA1 and TRPV1, by studying the effect of LPS on double *Trpa1/Trpv1* knockout mice. Our results demonstrate that genetic ablation of both channels slightly increases the Penh response, and has no statistically significant effect on leukocyte infiltration. On the other hand, pharmacological inhibition of TRPV4 in these double KO animals resulted in an abrupt increase in Penh and increase in cellular infiltration, supporting our previous data. These are included in Figure 7g and Supplementary Figures 11 and 14.

Reviewer #3 (Remarks to the Author):

Major concerns:

1. The biggest concern of this study is that the figures are not well organized which gives the manuscript a messy appearance. There are too many supplemental figures, most of which can be reorganized into a single figure by optimizing the size of the panels.

We thank very much Reviewer for the advice and for taking the time for finding solutions for the arrangements of the figures. We had many supplemental figures because we followed the principle of not mixing different experimental series, so that the reader would have less confusion. In our previous experiences other Reviewers have asked to separate figures. We also tended to keep out of the main figures non-essential examples and experiments with negative results. Nevertheless, we tried to comply with the present requests by changing the distribution of the figures as suggested.

- Sfigure 2b and Sfigure 3 should be combined with figure 1 as they are corresponding traces to the bar graphs in figure 1b and c.

Done, see panels 1c and 1g.

- The authors should also provide Ca^{2+} responding traces to match the 'WT+HC' bar in Figure 1b.

These traces are now provided in Figure 1g. As the Reviewer may notice we have added other panels to Figure 1 to make it more balanced in terms of space and distribution (see panels d and h).

- Sfigure 6 should be merged with Figure 3 to match Figure 3c and d.

Done, see panel 3g.

- In Figure 4, the authors should provide corresponding Ca^{2+} responding traces and NO responding traces to match panel c and d.

These traces are now provided in panels 4c and 4d and in the new Supplementary Fig. 3.

- Sfigure 8 should be included in Figure 5 to match Figure 5e. In addition, the authors should provide Ca^{2+} responding traces and NO responding traces to match e in Figure 5.

We moved previous Supplementary Fig. 9a to Figure 5d. Representative traces of intracellular Ca^{2+} and NO levels are now included in Figure 5d.

2. Since protein expression level of TRPM7 is relatively high compared with other TRP channels in both mTEC and 16HBE, why didn't the authors perform any experiments to rule

out the possibility of TRPM7 involvement in LPS-induced intracellular free Ca^{2+} increase? This study should provide evidence for this concern using TRPM7 activator, inhibitor or overexpression of TRPM7 in HEK293T as shown in Sfigure9 c in which TRPV3 was excluded from being activated by LPS.

Although not formally discussed in the text, evidence against a role of TRPM7 in the responses to LPS could be found throughout the paper. For instance, non-transfected HEK293T cells are known to have high expression of TRPM7, but are largely insensitive to LPS. Furthermore, TRPV4 inhibition, either genetically or pharmacologically, rendered epithelial cells largely insensitive to LPS. Nonetheless, we now demonstrate this point by showing that TRPM7 currents are not affected by LPS (see new Supplementary Fig. 7d, e).

3. The experiments of TRPV4-transfected HEK293T used to confirm the role of mouse TRPV4 in LPS-induced intracellular free Ca^{2+} increase in mouse TEC should be included with the supplemental materials.

The data on HEK239T cells transfected with the mouse TRPV4 add key descriptive and mechanistic details, such as the dose response curve for TRPV4 responses to LPS, the requirement of freely accessible lipid A moiety and the sensitivity to lamellar LPS (included in the revised version). Therefore, we would like to keep these results in a main figure.

4. In this study, Tlr4 pathway was excluded from the activation of TRPV4 by LPS. The authors provides sufficient evidence, however, the data organization is messy. The middle pane in Figure 1a, Figure 1b and Sfigure 6 should be moved to Figure 3 together with Tlr4 inhibitive data, which would make the figure or evidence appear uniform and integrated.

We agree that consolidating all data on TLR4 in a single figure could be one valid way of organizing the figures. However, we feel that the current distribution of figures is more suited for the flow of the text. Given that TLR4 is usually described as the sole receptor for LPS, we find it crucial to show from the very beginning of the manuscript that this pathway is not required for the LPS-induced Ca^{2+} responses. This allows us to concentrate on TRPV4, which is the main topic, and consequently show all other data on TLR4 as confirmatory. With all respect to this Reviewer's opinion, we would prefer keep the structure as it is proposed in the original version.

5. The third section of results mentioned that the LPS-induced increase in ciliary beat frequency is TRPV4 dependent. Can the author also include supplemental video data to support the summarized data shown in Sfigure 7?

In Supplementary Figure 7 (now Supplementary Figure 10) we show an average increase of 15% in the CBF of wild type mTEC exposed to LPS. Although small, this significant increase

is comparable to effects induced by 4αPDD, a known agonist of TRPV4 (Lorenzo IM et al., 2008. PNAS, 105(34):12611-6.). In these cells, basal CBF averages 10 Hz, which implies that a 15% increase corresponds to 11.5 Hz. For this reason, we use image-processing software capable of determining changes in beating frequencies otherwise imperceptible to the human eye. A representative video of this change would not bring further visual information, even in slow motion.

6. In Figure 5f, besides using TRPV4 inhibitor HC067047 to inhibit iNOS disaggregation, the authors should also use TRPV4 knockdown or knockout cells to further confirm the conclusion.

Figure 5f shows that the amount iNOS aggregates in 16HBE cells decrease after stimulation of TRPV4 either by LPS or GSK1016790A. To prove the role of TRPV4 in this effect we have used the well-known TRPV4 inhibitor HC067047. This compound clearly inhibits disaggregation of iNOS. The reviewer suggested that, in addition, we should provide experiments in which TRPV4 protein is knocked down. Although we agree that this would be an extra control, there are important reasons for which we respectfully disagree on including such data in this manuscript. First and foremost, these experiments would not bring new conceptual or translational value to the story. Second, we have no reason to doubt the specificity of HC067047 in the experiments on 16HBE cell, as this compound inhibits the responses to the specific TRPV4 agonist GSK1016790A. Finally, considering the constraints of time and resources we decided to concentrate our efforts on confirming the bactericidal effect of TRPV4-generated nitric oxide (Figure 6 in the revised version) and to obtain new data from primary cultured human nasal epithelial cells (Supplementary Fig. 4). We sincerely apologise for having to decline this request, in favour of the possibility of publishing the central message of our study without further delay.

Minor concerns:

1. In Figure 1b, please show standard deviation (SD) or standard error (SE) for each bar.

Former Figure 1b, now Figure 1i, shows absolute percentage of responders per condition gathered over different cell culture coverslips. Therefore, the bars on this graph do not have SD or SE. Accordingly, Fisher's exact test was used to compare the different populations. This is indicated in the figure legend.

2. The size of panels in each figure should be optimized to make the figures neat.

We now present figures with new panel distribution, and feel that they are of sufficient neatness for publication. We remain of course open for further suggestions for improvement from the Editorial office.

REVIEWERS' COMMENTS:

Reviewer #1 (Remarks to the Author):

Authors have adequately addressed my critiques. Acceptable for publication

Reviewer #2 (Remarks to the Author):

None

Reviewer #3 (Remarks to the Author):

No further comments.

TRPV4 activation triggers protective responses to bacterial LPS in airway epithelial cells. Alpizar Y. A. et al.

Answer to the Reviewers

We are very grateful to the reviewers for their critical evaluation of our work. Based on their final remarks, listed below, we have no further comments to add to the peer-reviewed process.

Reviewer #1 (Remarks to the Author):

Authors have adequately addressed my critiques. Acceptable for publication.

Reviewer #2 (Remarks to the Author):

None.

Reviewer #3 (Remarks to the Author):

No further comments.